# Identification and Characterization of High-Molecular-Weight Proteins Secreted by *Plasmodiophora brassicae* That Suppress Plant Immunity

**DOI:** 10.3390/jof10070462

**Published:** 2024-06-29

**Authors:** Yanqun Feng, Xiaoyue Yang, Gaolei Cai, Siting Wang, Pingu Liu, Yan Li, Wang Chen, Wei Li

**Affiliations:** 1MARA Key Laboratory of Sustainable Crop Production in the Middle Reaches of the Yangtze River (Co-Construction by Ministry and Province), College of Agriculture, Yangtze University, Jingzhou 434025, China; 17347479204@139.com (Y.F.); yangxy_1999@163.com (X.Y.); 18287369093@163.com (S.W.); 17872307535@163.com (P.L.); vgly1987@sina.com (Y.L.); 2Engineering Research Center of Ecology and Agricultural Use of Wetland, Ministry of Education, College of Agriculture, Yangtze University, Jingzhou 434025, China; 3Hubei Collaborative Innovation Center for Grain Industry, College of Agriculture, Yangtze University, Jingzhou 434025, China; 4Institute of Plant Protection, Jiangsu Academy of Agricultural Sciences, Nanjing 210014, China; 5Institute of Plant Protection, Shiyan Academy of Agricultural Sciences, Shiyan 442000, China; caigaolei@163.com

**Keywords:** clubroot disease, secreted protein, PTI, SA signaling pathway, JA signaling pathway

## Abstract

*Plasmodiophora brassicae* is an obligate intracellular parasitic protist that causes clubroot disease on cruciferous plants. So far, some low-molecular-weight secreted proteins from *P. brassicae* have been reported to play an important role in plant immunity regulation, but there are few reports on its high-molecular-weight secreted proteins. In this study, 35 putative high-molecular-weight secreted proteins (>300 amino acids) of *P. brassicae* (PbHMWSP) genes that are highly expressed during the infection stage were identified using transcriptome analysis and bioinformatics prediction. Then, the secretory activity of 30 putative PbHMWSPs was confirmed using the yeast signal sequence trap system. Furthermore, the genes encoding 24 PbHMWSPs were successfully cloned and their functions in plant immunity were studied. The results showed that ten PbHMWSPs could inhibit flg22-induced reactive oxygen burst, and ten PbHMWSPs significantly inhibited the expression of the SA signaling pathway marker gene *PR1a*. In addition, nine PbHMWSPs could inhibit the expression of a marker gene of the JA signaling pathway. Therefore, a total of 19 of the 24 tested PbHMWSPs played roles in suppressing the immune response of plants. Of these, it is worth noting that PbHMWSP34 can inhibit the expression of JA, ET, and several SA signaling pathway marker genes. The present study is the first to report the function of the high-molecular-weight secreted proteins of *P. brassicae* in plant immunity, which will enrich the theory of interaction mechanisms between the pathogens and plants.

## 1. Introduction

To resist pathogen infection, plants employ a two-layered immune system composed of pathogen-associated molecular pattern (PAMP)-triggered immunity (PTI) and effector-triggered immunity (ETI) [1,2]. PTI is triggered by the recognition of PAMPs by pattern recognition receptors, which are the first level of immune response that can prevent the colonization of most nonpathogenic microorganisms. However, plant pathogens can secrete effectors into plant cells to interfere with plant PTI and successfully infect the host [1,2]. In view of their important roles in the interactions between plant pathogens and plants, the effector proteins of many plant pathogens have been extensively studied [3,4,5,6].

In eukaryotic pathogens, as with plant pathogenic fungi and oomycetes, effector proteins are first secreted into the apoplast. Some effector proteins, like RxLR effector proteins, are able to enter cells and target the host components to regulate plant immunity [7,8,9]. Usually, effector proteins are defined as small secreted proteins that are typically smaller than 300 amino acids (aa) (the approximate molecular weight) [3,10,11,12]. However, in addition to small secreted proteins, eukaryotic pathogens also encode many high-molecular-weight secreted proteins (HMWSPs) [10,12], which have only been the focus of a few studies.

Clubroot, caused by *Plasmodiophora brassicae* (*P*. *brassicae*), is a soil-borne disease that affects cruciferous crops worldwide [13,14,15], reported in more than 80 countries [16]. *P. brassicae* is an intracellular obligate biotrophic protozoan. During its life cycle, except for the resting spore and zoospore stages, *P. brassicae* always inhabits plant cells [17]. Consequently, in theory, all of the proteins secreted by *P. brassicae* are located in host cells during infection, and these proteins may act as effectors. However, until now, only the small secreted proteins of *P. brassicae* have been reported to be effector candidates.

The genome of *P. brassicae* was first sequenced by Schwelm et al. [18]. In this research, only proteins smaller than 450 aa were identified as putative secreted proteins. In the research on the identification of effector candidates during *P. brassicae* infection, only secreted proteins smaller than 300 or 400 aa were analyzed in primary or secondary infection studies, respectively [19,20]. It was found that *P. brassicae* encoded a large number of high-molecular-weight secreted proteins. An analysis of a draft genome of *P. brassicae* pathotype 3 showed that 590 secreted proteins were identified as putative secreted proteins, of which only 221 were smaller than 300 aa [21]. Furthermore, *P. brassicae* encodes a secreted methyltransferase, PbBSMT, that can methylate SA (salicylic acid is an important endogenous signal molecule in the activation of plant defense responses) and further alter host susceptibility The length of PbBSMT is 378 aa, and its mature protein has a length of 357 aa [22,23,24]. These reports suggest that some high-molecular-weight secreted proteins may also play a vital role during interactions between *P. brassicae* and its host. However, there is still no systematic report about the high-molecular-weight secreted proteins of *P. brassicae*.

In the present research, the putative HMWSP (>300 aa) genes from the *P. brassicae* e3 genome were systematically identified. The transcriptome data from the different infection stages of *P. brassicae* were analyzed, and the genes encoding HMWSPs that were highly expressed during the intracellular parasitic stage were identified and cloned. Furthermore, the secretory activities of these putative high-molecular-weight secreted proteins were determined by the yeast signal sequence trap system, and the functions of these secreted proteins in the PTI response were investigated. Finally, the effects of these high-molecular-weight secreted proteins on the salicylic acid (SA), jasmonic acid (JA), and ethylene (ET) signaling pathways were also quantified. This work identified the high-molecular-weight secreted proteins of *P. brassicae* and revealed their functions in plant immunity, which will lay the foundation for further revealing the function mechanisms of *P. brassicae*’s high-molecular-weight secreted proteins and broadening our understanding of them.

## 2. Materials and Methods

### 2.1. P. brassicae and Plant Materials

The *P. brassicae* isolate used in the present study was collected from a *Brassica napus* (*B. napus*) field located in Zhijiang, Yichang City, Hubei Province, China. The clubroot gall was stored at −80 °C. To inoculate *B. napus*, the thawed clubroot was homogenized in a blender with an approximately 10-fold higher volume of distilled water, and the grinding mixture was filtered through eight layers of cheesecloth. The filtrated resting spore suspension was adjusted to a concentration of 1 × 10^6^ (spores)/mL and then used to inoculate 7-day-old *B. napus* seedlings. *Nicotiana benthamiana* (*N. benthamiana*) and *B. napus* were grown in a greenhouse at 22 °C with a 16 light/8 h dark cycle.

### 2.2. Prediction of High-Molecular-Weight Secreted Protein Genes That Are Specifically Highly Expressed during Infection

The *P. brassicae* e3 genome (Accession NO. LS992577) sequenced by PacBio RSII sequencing technology [25] was used for predicting putative HMWSPs (>300 aa). The signal peptides of all protein sequences from *P. brassicae* e3 were predicted using SignalP v6.0 [26]. The transmembrane regions were predicted using TMHMM v2.0 [27]. Only the proteins containing a signal peptide, but not an extra transmembrane region, were identified as putative secreted proteins. The proteins that were larger than 300 aa were defined as high-molecular-weight secreted proteins.

To obtain the putative HMWSPs specifically highly expressed during the intracellular parasitic stage, two series of transcriptome data were analyzed. One was obtained from *P. brassicae* single-spore isolate e3 resting spores and infected plant samples (SRA Accession NO. ERX1409401, ERX1409399, ERX1409398, and ERX1409397). The other was obtained from the resting spores and infected plant samples of the *P. brassicae* strain ZJ-1 (SRA Accession NO. SRS4029411, SRS4029410, SRS4029409, SRS4029408, SRS4029407, SRS4029406, SRS4029405, SRS4029404, and SRS4029403). To quantify the expression level of the HMWSP genes, the transcriptome data were mapped to the *P. brassicae* e3 genome using HISAT2 [28], and then the expression levels of the *P. brassicae* genes were calculated with StringTie (normalized as FPKM value) [29]. If the expression level of a *P. brassicae* gene fit any of the following conditions, the gene was defined as a putative HMWSP specifically highly expressed during the intracellular parasitic stage: (1) the FPKM value was 0 in resting spores and greater than 0 in infected plant tissues; or (2) the FPKM value in infected plant tissues was 10 times that in resting spores. Finally, the putative HMWSPs specifically highly expressed during the intracellular parasitic stage identified in both series of transcriptome data were used for further research.

### 2.3. Functional Evaluation of the Signal Peptides of Putative HMWSPs from P. brassicae

The secretory activity of the putative HMWSPs was evaluated using a yeast signal sequence trap system, as previously reported [30,31]. The fragments encoding the signal peptides of the putative HMWSPs were amplified and cloned into the yeast signal peptide trap vector pSUC2T7M13ORI using ClonExpress II One Step Cloning kit (Vazyme, Nanjing, China). The recombinant plasmid was verified with sequencing and was transformed into the invertase negative yeast strain YTK12. The transformants were grown on CMD-W medium (0.67% yeast N base without amino acids, 0.075% tryptophan dropout supplement, 2% sucrose, 0.1% glucose, 2% agar, pH 5.8). The transformants, confirmed with PCR, were used for a functional evaluation of the signal peptide using a growth assay on YPRAA media (1% yeast extract, 2% peptone, 2% raffinose, and 2 µg/mL of antimycin A) and a 2,3,5-triphenyltetrazolium chloride (TTC) assay. Only transformants carrying functional signal peptides can grow normally on YPRAA medium and reduce the transparent TTC to the insoluble red-colored 1,3,5-triphenylformazan [32]. A fragment encoding a functional signal peptide from Ps87 of *Puccinia striiformis* was used as a positive control, and a fragment encoding a non-functional signal peptide of Mg87 from *Magnaporthe oryzae* was used as a negative control [33]. The primers used for amplifying the fragments encoding the signal peptides of the predicted HMWSPs are listed in Appendix A.

### 2.4. RNA Extraction and cDNA Synthesis

Approximately 100 mg of collected plant leaves or root samples were used for RNA extraction using the TRIzol reagent (Invitrogen, Carlsbad, CA, USA) following the manufacturer’s instructions. Then, 1 µg of total RNA was used to synthesize the cDNA using a PrimeScript™ RT reagent kit with gDNA Eraser (Takara, Dalian, China) according to the manufacturer’s protocol.

### 2.5. Expression of HMWSPs of P. brassicae in N. benthamiana

Twenty days after infection with *P. brassicae*, the *B. napus* roots were collected for RNA extraction and the subsequent cDNA synthesis. Then, the cDNA was used to amplify the fragments encoding the mature secreted proteins using the high-fidelity enzyme I5 (Tsingke, Beijing, China). The primers are shown in Appendix A. The fragment encoding the mature secreted protein was transformed into the plant expression vector pBI121flag, and the recombinant plasmid was transformed into *Agrobacterium tumefaciens* (*A. tumefaciens*) GV3101 using electroporation. The *A. tumefaciens* verified with PCR was re-suspended twice using a suspension buffer (10 mM MgCl_2_, 10 mM MES pH 5.8, 200 μM AS). After dilution to an OD600 of 1, the *A. tumefaciens* suspension was injected into the leaves of *N. benthamiana* at the 5–6-leaf stage [34]. Forty-eight hours later, the *N. benthamiana* leaves were collected. For the detection of the expression of HMWSP genes in *N. benthamiana*, the total RNA of *N. benthamiana* inoculated with *A. tumefaciens* was extracted. After erasing the gDNA, the first-strand cDNA was synthesized, and then the expression of the HMWSP genes was detected with qRT-PCR. The primers used in the qRT-PCR assay are listed in Appendix A.

### 2.6. Flg22-Induced Reactive Oxidative Species (ROS) Burst Assay

The Flg22-induced ROS burst in *N. benthamiana* leaves was quantified using the luminol chemiluminescence method with minor modification [35]. Forty-eight hours post *A. tumefaciens* inoculation, the *N. benthamiana* leaf discs were collected and placed in a Petri dish containing deionized water in the dark for 2h (changing the deionized water every 30 min), and then maintained in the dark overnight in a 96-well luminometer plate with 200 μL water in each well. The next day, the distilled water in the microplate was replaced with 200 μL of 1.25 × luminol/horseradish peroxidase working solution (0.0375 mg/mL luminol, 0.025 mg/mL horseradish peroxidase), which the leaf discs were kept completely submerged in. After adding 50 μL of 5 × flg22 solution (500 nM flg22), the chemiluminescence signal was measured immediately for 40 min with a signal integration time of 1 sec using a Multimode Reader [36]. Three *N. benthamiana* leaves were injected with each gene, and 12 leaf discs were taken for flg22-induced ROS burst detection. The maximum chemiluminescence signal value of each leaf disc was recorded over 40 min, and Student’s *t*-test was used to determine significant (*p* < 0.05) or extremely significant (*p* < 0.01) differences between the experimental and control groups.

### 2.7. Quantification of the Expression Levels of Marker Genes of the SA, JA, and ET Signaling Pathways

To investigate the roles of the HMWSPs participating in the SA-, JA-, and ET-mediated disease resistance pathways, the expression levels of marker genes related to SA, JA, and ET in *N. benthamiana* were quantified using qRT-PCR. The selected marker genes for the SA-mediated disease resistance pathway were *PR1a*, *PR1b*, *PR2b*, *PR5*, and *NPR1*. The marker genes selected for the JA-mediated disease resistance pathway were *LOX* and *PDF1.2* [37,38,39,40], and the marker gene selected for the ET signaling pathway was ERF [38]. The *EF-1α* gene was used as the internal reference [38]. The primer information is shown in Appendix A. The *N. benthamiana* leaves at 48 h after *A. tumefaciens* injection were used for RNA extraction, cDNA synthesis, and a subsequent qRT-PCR analysis. Each 20 µL qRT-PCR system contained 10 µL of 2 × quantitative SYBR mix (Vazyme, Nanjing, China), 10 µM of forward and reverse primers, 2 µL of cDNA, and 8.7 µL of RNase-Free ddH_2_O. The procedure of quantitative PCR was 94 °C for 30 s; 59 °C for 20 s; 72 °C for 30 s. For each treatment, three biological replicates and three technical repeats were carried out, and the *N. benthamiana* leaves injected with *A. tumefaciens* containing an empty vector were used as controls. The relative expression levels of genes were calculated using the 2^−∆∆CT^ method [41].

### 2.8. Statistical Analyses

Statistical significances based on the *t*-test were determined with Prism 7.0 software (GraphPad Software). The Student’s *t*-test was used to determine significant (*p* < 0.05) or extremely significant (*p* < 0.01) differences between the experimental and the control groups.

## 3. Results

### 3.1. P. brassicae Encodes a Large Number of Putative High-Molecular-Weight Secreted Proteins

In order to identify the putative PbHMWSPs, a total of 9230 proteins encoded by the *P. brassicae* e3 genome were analyzed. After signal peptide prediction using SignalP V6.0, 1086 of the proteins were found to contain a signal peptide. The transmembrane region prediction of these 1086 proteins was performed with TMHMM V2.0. The results showed that 709 proteins did not contain extra transmembrane regions, and they were identified as putative secreted proteins (Figure 1A). Furthermore, the length of these putative secreted proteins was analyzed. In total, 407 secreted proteins with a length greater than 300 amino acids were identified as putative PbHMWSPs, accounting for more than half (57.4%) of the number of putative secreted proteins (Figure 1B). 

To obtain the PbHMWSPs that were highly expressed during the intracellular parasitic stage, the transcriptome data of the *P. brassicae* resting spores and infected samples were analyzed. The transcriptome analysis showed that 92, 75, 59, and 86 putative PbHMWSP genes were highly expressed in e3-infected *B. rapa*, *B. napus*, and *B. oleracea* and in ZJ-1-infected *B. napus,* respectively. Based on the above results, 35 putative PbHMWSP genes were highly expressed in all four samples (Figure 1C, Appendix A). Consequently, these 35 putative PbHMWSPs, which may play important roles in *P. brassicae* infection, were used in the subsequent studies. 

In order to further understand the characteristics of these 35 putative PbHMWSPs, their functional domains were analyzed. Among these 35 putative PbHMWSPs, 6 putative PbHMWSPs (PbHMWSP14, PbHMWSP21, PbHMWSP22, PbHMWSP26, PbHMWSP28, and PbHMWSP35) had no known domains, and the remaining 29 PbHMWSPs possessed known domains such as lipase, benzoic acid/salicylic acid methyltransferase, and protein kinases.

### 3.2. Thirty High-Molecular-Weight Secreted Proteins Showed Secretory Activity

To confirm our signal peptide prediction results, the 35 putative PbHMWSPs were further investigated using a yeast signal sequence trap system to determine the secretory activities of their signal peptide sequences [42]. The results showed that the yeast strains containing the signal peptide sequences of 29 putative PbHMWSPs grew normally on YPRAA plates and catalyzed TTC into insoluble red TPF, similar to the positive controls (yeast strains containing Ps87) (Figure 2 and Appendix A). However, similar to the negative controls (yeast strains containing Mg87), the yeast strains containing the signal sequences of the remaining five putative PbHMWSP peptides grew slowly on the YPRAA plate and could not catalyze the colorless TTC solution into insoluble red TPF, the solution of which was still transparent (Figure 2 and Appendix A). Consequently, these five putative PbHMWSPs were not considered to have secretory activity. In addition, the signal peptide from PbHMWSP1, SPQ93076.1 (called PbBSMT), has been previously reported to be functional [23]. These 30 putative PbHMWSPs were validated to have secretory activity and were used for further research in the present study.

### 3.3. Ten High-Molecular-Weight Secreted Proteins Suppress the flg22-Induced ROS Burst in N. benthamiana

In order to study the roles of the above 30 PbHMWSPs in plant PTI, the fragments encoding the mature proteins (without signal peptides) of these PbHMWSPs were cloned (the gene sequences of the PbHMWSPs are shown in Appendix A), and 24 of them were obtained. Then, the fragments encoding mature proteins were cloned into the plant expression vector pBI121flag and expressed in *N. benthamiana* by the *A. tumefaciens*-mediated transient expression system. RT-PCR was used to detect the expression level of these 24 PbHMWSPs, which showed that all of them were successfully expressed (Appendix A). After treatment with flg22, the reactive oxygen burst of the *N. benthamiana* leaves expressing PbHMWSPs was quantified. The *N. benthamiana* leaves expressing GUS were used as the controls. We first evaluated the effect of GUS on an flg22-triggered induced ROS burst; the max relative luminescence unit (RLU) values of *N. benthamiana* leaves expressing GUS were not significantly different from *N. benthamiana* leaves injected with either *A. tumefaciens* or with *A. tumefaciens* carrying the empty vector pBI121flag. This indicated that GUS expression did not affect the flg22-triggered ROS burst in *N. benthamiana* leaves. Compared with the *N. benthamiana* leaves expressing GUS, the maximum values of the relative luminescence units (RLUs) of *N. benthamiana* leaves that expressed ten PbHMWSPs (numbered 6, 8, 11, 12, 13, 16, 17, 20, 22, and 35) were significantly lower, while the maximum RLU values of *N. benthamiana* leaves expressing nine PbHMWSPs (numbered 1, 3, 10, 19, 25, 27, 28, 31, and 34) were significantly higher than that of the controls and the maximum RLU values of the remaining five PbHMWSPs were not significantly different to those of the controls (Figure 3). These results showed that the ROS burst was significantly suppressed in *N. benthamiana* leaves expressing ten PbHMWSPs and was induced in *N. benthamiana* leaves expressing nine PbHMWSPs. These results indicated that ten PbHMWSPs (numbered 6, 8, 11, 12, 13, 16, 17, 20, 22, and 35) could suppress flg22-mediated PTI.

### 3.4. Effect of High-Molecular-Weight Secreted Proteins on SA-Mediated Disease Resistance Signaling Pathway

The SA-mediated signaling pathway plays an important role in plant resistance to biotrophic pathogens [43,44,45]. *P. brassicae* is a typical biotrophic parasite that has been reported to be resisted by the SA pathway in plants [46,47,48]. To investigate the roles of PbHMWSPs on the SA signaling pathway, the expression levels of SA pathway marker genes in *N. benthamiana* leaves were quantified with heterologously expressed PbHMWSPs. As shown in Figure 4A, compared with the controls, the relative expression of PR1a levels was significantly decreased in *N. benthamiana* leaves expressing ten PbHMWSPs (numbered 1, 3, 13, 16, 22, 25, 26, 28, 34, and 35). The reductions in the relative expression of *PR1a* ranged from 21.2% to 81.3% (Table 1). In addition, the relative expression levels of *PR1a* in *N. benthamiana* leaves expressing seven PbHMWSPs (numbered 6, 12, 15, 19, 20, 24, and 31) were significantly up-regulated compared with the control, and the increases ranged from 78.3% to 411%. The ten PbHMWSPs suppressing the expression of *PR1a* were used for further qPCR analysis to quantify the expression levels of the other marker genes of the SA pathway (*PR1b*, *PR2b*, and *PR5*). Compared with the controls, the expression levels of *PR1b*, *PR2b*, and *PR5* were all down-regulated only in *N. benthamiana* leaves expressing two PbHMWSPs (PbHMWSP1 and PbHMWSP34) (Figure 4B). More strikingly, in *N. benthamiana* leaves expressing PbHMWSP34, the expression levels of *PR1b*, *PR2b*, and *PR5* were no more than 2% of that of the control.

### 3.5. Effect of High-Molecular-Weight Secreted Proteins on JA Disease Resistance Signaling Pathway

The JA pathway plays an important role in plant resistance to necrotrophic pathogens [44,45]. Although *P. brassicae* is a typical biotrophic pathogen, the research has shown that JA also plays an important role in plant resistance to clubroot disease [46,51,52]. To investigate the effect of PbHMWSPs on the JA signaling pathway, the expression levels of JA pathway marker genes were quantified in *N. benthamiana* leaves’ heterologously expressed PbHMWSPs (Figure 5). Two JA pathway marker genes, *PDF1.2* and *LOX*, were selected. The expression levels of *PDF1.2* were extremely low in all samples, and so only the expression levels of *LOX* were further analyzed. The results showed that nine PbHMWSPs (numbered 3, 8, 17, 25, 27, 28, 31, 33, and 34) could significantly inhibit the expression of LOX. Among them, PbHMWSP25 showed the most significant inhibition of *LOX* expression, of which the expression level in *N. benthamiana* leaves expressing PbHMWSP25 was only 41.5% of that of the controls (Table 1). Thirteen PbHMWSPs (numbered 1, 6, 7, 10, 11, 12, 15, 16, 19, 20, 22, 24, and 26) significantly induced the expression of the JA resistance pathway marker gene *LOX*, among which the highest expression level was found in *N. benthamiana* leaves expressing PbHMWSP6, which was 78.3 times that of the controls. This suggested that *P. brassicae* PbHMWSPs were also involved in the regulation of the JA signaling pathway.

### 3.6. Effect of High-Molecular-Weight Secreted Proteins on ET Disease Resistance Signaling Pathways

In plants, ET participates in the defense of necrotrophic pathogens together with the JA disease resistance pathway [53,54,55]. It has been reported that ET is a positive regulator against *P. brassicae*, and it activates the SA signaling pathway and transcription factors including WRKY75, WRKY45, SHN1, and At2G20350, thereby delaying the primary infection [56]. In order to study the effect of PbHMWSPs on the ET signaling pathway, the expression levels of the ET signaling pathway marker gene *ERF* were quantified in *N. benthamiana* leaves that heterologously expressed PbHMWSPs (Figure 6). The results showed that seven PbHMWSPs (numbered 1, 8, 13, 22, 25, 26, and 34) could significantly suppress the expression of the ET pathway marker gene ERF, among which the most significant inhibition was caused by PbHMMSP26. The expression level of ERF in *N. benthamiana* leaves that heterologously expressed PbHMWSP26 was only 19.7% of that of the controls (Table 1). In addition, ten PbHMWSPs significantly induced the expression of *ERF*, among which the highest increase was caused by PbHMWSP19. The expression levels of *ERF* in *N. benthamiana* leaves that heterologously expressed PbHMWSP19 were 23.7 times of that of the controls. These results showed that PbHMWSPs also participate in the regulation of the ET pathway, showing varying effects.

## 4. Discussion

The effector proteins secreted by plant pathogens play important roles in plant–pathogen interactions [3,4,5,6,57]. Some small molecular proteins secreted (<300 aa) by pathogenic fungi and oomycetes can be transported into plant cells and target host proteins to promote infection, which have become a hot spot in the study of plant pathogens [9,33,58]. Similarly, the small molecules of secreted proteins have been the focus in studies of interactions between *P. brassicae* and plants [19,59,60]; in contrast, the present study, for the first time, identified high-molecular-weight secreted proteins of *P. brassicae* and found that they also had the ability to inhibit plant immunity, laying the foundation for further study of the functional mechanisms of these proteins and enriching the theory of plant–pathogen interactions.

Flg22, a small peptide that can induce PAMP-triggered immunity in plants, was widely used to evaluated the function of secreted proteins from plant pathogens on plant immunity [61]. In this study, ten PbHMWSPs could suppress flg22-mediated PTI. In the previous work, the inhibition of plant immunity by other *P. brassicae* secreted proteins has also been reported. Zhan and collaborators analyzed the functions of 63 secreted proteins (less than 450 aa) from *P. brassicae* related to plant immunity, and found that 55 of the secreted proteins could inhibit BAX-induced cell necrosis [62]. Chen and his group evaluated the effects of 33 secreted proteins (less than 300 aa) of *P. brassicae* on plant immunity, and found that 24 of the secreted proteins could inhibit BAX-induced cell necrosis [20]. Hossain and other researchers [59] determined the PTI functions of 12 *P. brassicae* proteins located in the intimal system, and found that 7 of them could inhibit INF1- and NPP1-induced cell necrosis. These reports showed that a large number of the *P. brassicae* secreted proteins had the ability to suppress plant immunity, whether they were PbHMWSPs or other secreted proteins. This may be due to the intracellular parasitism of *P. brassicae*. *P. brassicae* needs a large number of secreted proteins to overcome plant immunity.

The previous research showed that the SA, JA, and ET signaling pathways played important roles in plant resistance to *P. brassicae* [47,48,49,52,53,57]. However, how these signaling pathways are suppressed by *P. brassicae* is still not well understood. Until now, only one *P. brassicae* effector protein PbBSMT was reported to inhibit SA signaling pathways through methylate SA to MeSA [22,23,24]. Fortunately, PbBSMT was also identified in the present study, which was numbered as PbHMWSP1. Furthermore, we also found that the overexpression of PbHMWSP1could suppress the expression of SA marker genes. Except for PbHMWSP1, the expression levels of SA, JA, or ET marker genes were down-regulated in *N. benthamiana* of overexpressing another 14 PbHMWSPs as well. Consequently, we speculated that these 14 PbHMWSPs are probably able to regulate the SA, JA or ET signaling pathways, which still needs to be further verified.

Many proteins secreted by fungi do not contain any known domains [3]. In this paper, six PbHMWSPs did not contain any known domains, while the other twenty-four PbHMWSPs did. These known domains give us some clues for exploring these PbHMWSPs; for example, PbHMWSP1 contains benzoic acid/salicylic acid methyltransferase, which has been reported to catalyze salicylic acid into methyl salicylate, thereby reducing the content of SA and promoting the pathogenesis of *P. brassicae* [22,24]. Both PbHMWSP3 and PbHMWSP16 contain a lipase domain; PbHMWSP3 contains a Lipase_3 domain, while PbHMWSP16 contains an Abhydro_lipase domain. According to genomics, *P. brassicae* encodes 295 lipid-droplet-associated proteins, which are mainly linked to biosynthesis and metabolism [63]. Whether these two PbHMWSPs participate in the metabolism of the lipid droplets remains to be further studied. Furthermore, excess fatty acids were correlated with the formation of JA [64,65]. The expression level of the JA signaling pathway marker gene LOX in PbHMWSP3-expressing *N. benthamiana* leaves was extremely down-regulated, only 9% of that in the control. Whether this was correlated with its Lipase_3 domain is also worth further study. PbHMWSP33 and PbHMWSP34 contain the LRR domain, which is a key recognition site for many innate immune receptors [66]. PbHMWSP33 can only inhibit the JA signaling pathway, while PbHMWSP34 can significantly inhibit the SA, JA, and ET signaling pathways. Therefore, the two LRR PbHMWSP may have different functions in plant immune regulation. Meanwhile, it is worth noting that PbHMWSP34 is the only secreted protein that can inhibit the expression of all SA signaling pathway marker genes, and its comprehensive inhibitory effect is even better than PbHMWSP1 (a methyltransferase), which can methylate SA and further negatively regulate the SA signaling pathway. PbHMWSP13 and PbHMWSP31 contain protein kinase domains; kinase plays an important role in plant growth and development, stress, and immunity [67,68]. Some researchers have shown that pathogens also secrete some kinase effectors to manipulate these signaling pathways and promote disease [57,69]. In addition, PbHMWSP6 contains a zinc finger structure. PBCN_001987, an effector protein of *P. brassicae* identified by Chen et al. [20], also contains a zinc finger structure, which can inhibit the function of SnRK1.1 in plants and promote disease. All in all, these domains provide clues for understanding the functions and mechanisms of these proteins.

In summary, this study demonstrated the functions of *P. brassicae* PbHMWSPs in regulating plant immunity, enriched the theory of interaction between *P. brassicae* and plants, and also provided clues to reveal the mechanisms of action of these high-molecular-weight effector proteins.

## Figures and Tables

**Figure 1 jof-10-00462-f001:**
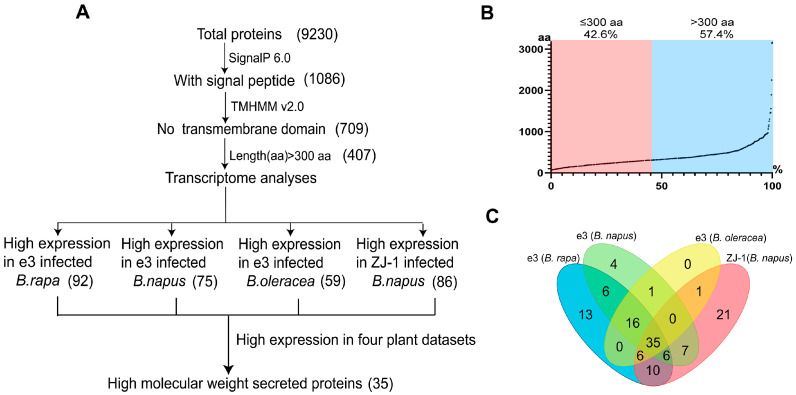
Prediction process and distribution of high-molecular-weight secreted proteins in *P. brassicae*. (**A**) Pipeline for the prediction of *P. brassicae* high-molecular-weight secreted proteins; numbers in parentheses indicate the number of proteins identified in each step. (**B**) Distribution of amino acids of putative secreted proteins. (**C**) Venn diagram of high-molecular-weight secreted proteins predicted to be highly expressed in e3^10^-infected *B. rapa*, *B. napus*, and *B. oleracea* and ZJ-1 infected *B. napus*.

**Figure 2 jof-10-00462-f002:**
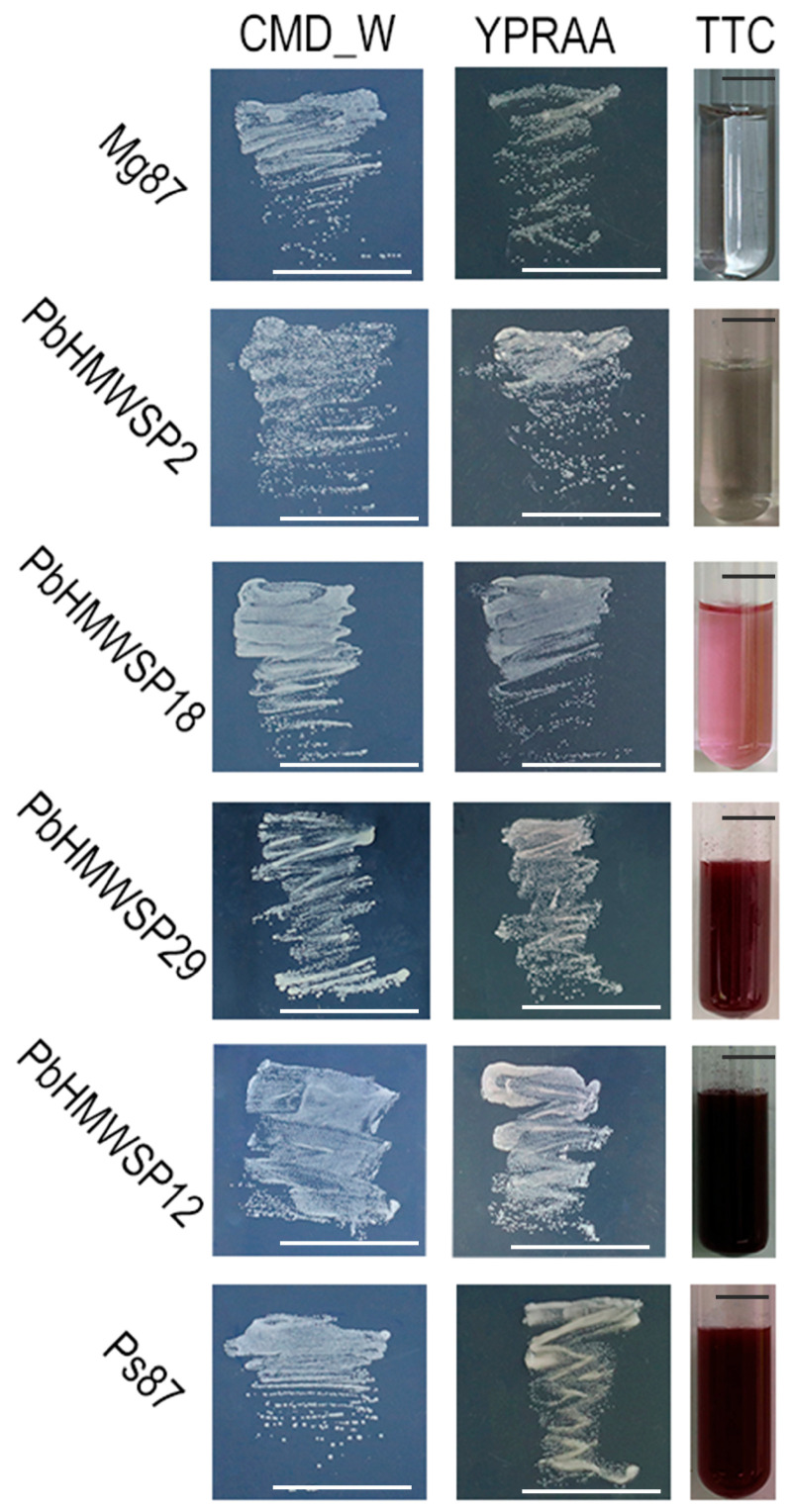
Functional validation of the signal peptides of high-molecular-weight secreted proteins from *P. brassicae* using a yeast signal sequence trap. The growth of the YTK12 strain carrying the putative secreted proteins PbHMWSP2, PbHMWSP18, PbHMWSP29, and PbHMWSP12 on YPRAA plates and the reactions of the proteins to the TTC assay were listed. The yeast strains containing the PbHMWSP12 or PbHMWSP29 signal peptide sequences grew normally on YPRAA plates and catalyzed TTC into insoluble red TPF, which indicated their secretory activity. The yeast strains containing the PbHMWSP2 or PbHMWSP18 peptide sequences grew slowly on the YPRAA medium and could not catalyze TTC into insoluble red TPF, which indicated that PbHMWSP2 and PbHMWSP18 did not have secretory activity. The *Magnaporthe oryzae* (Mg87) and *Puccinia striiformis* f. sp *tritici* (Ps87) signal peptides were used as negative and positive controls, respectively. Bars = 1 cm.

**Figure 3 jof-10-00462-f003:**
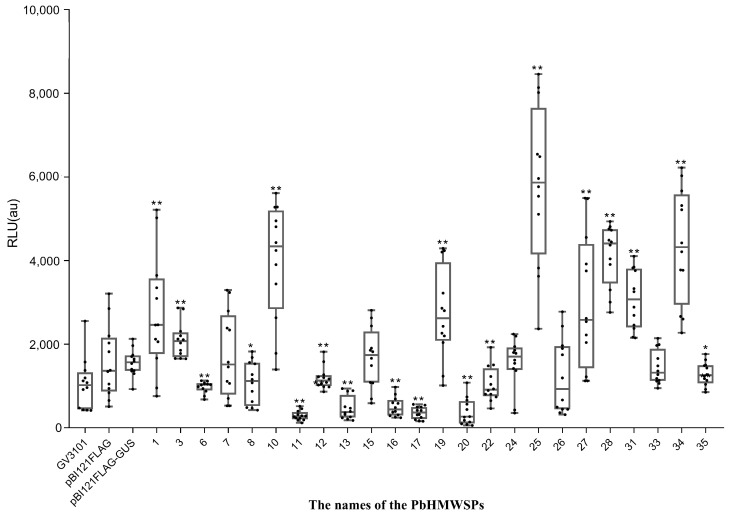
Flg22-induced reactive oxygen species (ROS) burst in *N. benthamiana* leaves overexpressing the high-molecular-weight secreted proteins and controls. Boxes extend from the 25th to the 75th percentile; whiskers extend from the lowest to highest values; bars indicate the median; *n* = 16 leaf discs. Statistically significant differences to the flg22-treated empty vector controls are indicated (Student’s *t*-test: *, *p* < 0.05; **, *p* < 0.01).

**Figure 4 jof-10-00462-f004:**
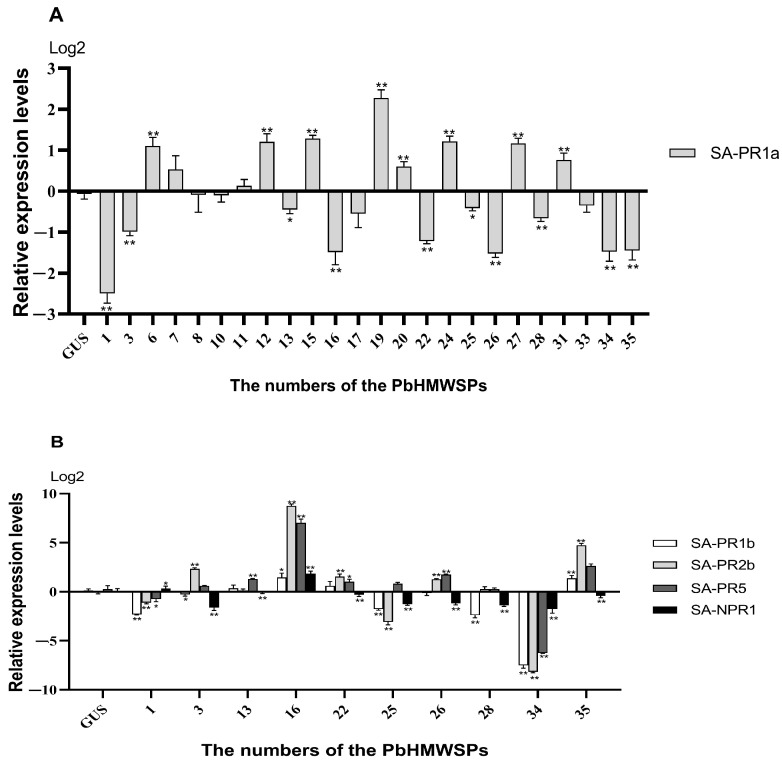
(**A**) The relative expression of SA pathway marker genes (*PR1a*) after treatment with high-molecular-weight secretory proteins. (**B**) The relative expression of SA pathway marker genes (*PR1a*, *PR1b*, *PR2b*, *PR5*, *NPR1*) after treatment with high-molecular-weight secretory proteins. The y axis represents the relative expression levels, and the x axis shows the numbers of the PbHMWSPs (high-molecular-weight secreted proteins). The data were analyzed with three independent repeated analyses, and the standard deviation is shown by the error bar. The GUS candidate effector was an empty vector without a flag fragment and served as a control. Statistically significant differences to the GUS are indicated (Student’s *t*-test: *, *p* < 0.05; **, *p* < 0.01).

**Figure 5 jof-10-00462-f005:**
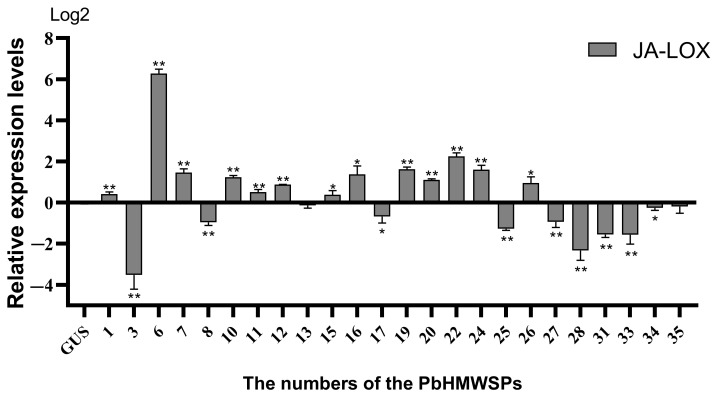
The relative expression levels of the JA pathway marker gene *LOX* after treatment with high-molecular-weight secretory proteins. The y axis represents the relative expression levels, and the x axis shows the numbers of the high-molecular-weight secreted proteins (PbHMWSPs). The data were analyzed with three independent repeated analyses, and the standard deviation is shown by the error bar. The GUS candidate effector was an empty vector without a flag fragment and served as a control. Statistically significant differences to the GUS are indicated (Student’s *t*-test: *, *p* < 0.05; **, *p* < 0.01).

**Figure 6 jof-10-00462-f006:**
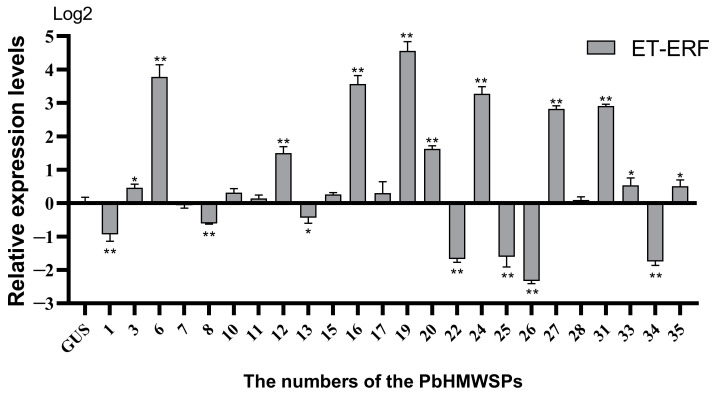
The relative expression of the ET pathway marker gene *ERF* after treatment with high-molecular-weight secretory proteins. The y axis represents the relative expression levels, and the x axis shows the numbers of the high-molecular-weight secreted proteins (PbHMWSPs). The data were analyzed with three independent repeated analyses, and the standard deviation is shown by the error bar. The GUS candidate effector was an empty vector without a flag fragment and served as a control. Statistically significant differences to the GUS are indicated (Student’s *t*-test: *, *p* < 0.05; **, *p* < 0.01).

**Table 1 jof-10-00462-t001:** Details of putative high-molecular-weight secreted proteins.

Name	e3 Homolog	Protein Size	Identity (%)	Functional Domains	Functional SignalP	Inhibition flg22 Induced ROS	Relative Control Expression of Marker Gene (%)						
							*PR1a*	*NPR1*	*PR1b*	*PR2b*	*PR5*	*LOX*	*ERF*
PbHMPSP1	SPQ93076	356	100	Methyltransf_7	Yes	No	18.7 ± 2.8	122.3 ± 22.9	17.9 ± 0.4	46.5 ± 3.8	48.9 ± 11.2	132.9 ± 9.9	52.3 ± 7.5
PbHMPSP2	SPQ93077	825	100	TRF4	No	—	—	—	—	—	—	—	—
PbHMPSP3	SPQ93093	633	100	Lipase_3	Yes	No	52.8 ± 3.3	31.9 ± 6.4	75.3 ± 10.5	501.8 ± 46.7	123.6 ± 4.4	9.3 ± 4.2	136.8 ± 10.3
PbHMPSP4	SPQ93860	507	100	FAD_binding_4	Yes	—	—	—	—	—	—	—	—
PbHMPSP5	SPQ94083	691	100	FAA1	Yes	—	—	—	—	—	—	—	—
PbHMPSP6	SPQ95067	412	100	RING-finger	Yes	Yes	226.8 ± 30.8	—	—	—	—	7831.0 ± 1214.0	1394.1 ± 367.6
PbHMPSP7	SPQ95194	373	95.642	Asp	Yes	No	154.2 ± 36.1	—	—	—	—	276.0 ± 35.0	94.8 ± 5.6
PbHMPSP8	SPQ95233	423	100	Asp	Yes	Yes	101.2 ± 25.6	—	—	—	—	51.5 ± 5.3	65.1 ± 0.9
PbHMPSP9	SPQ95590	496	99.033	PTZ00049	No	—	—	—	—	—	—	—	—
PbHMPSP10	SPQ95766	347	100	S_TKc	Yes	No	98.1 ± 10.8	—	—	—	—	235.6 ± 14.5	123.8 ± 10.1
PbHMPSP11	SPQ95794	415	96.087	ANK	Yes	Yes	115.1 ± 12.2	—	—	—	—	142.9 ± 12.0	109.7 ± 7.3
PbHMPSP12	SPQ95797	374	100	BTB	Yes	Yes	243.4 ± 30.6	—	—	—	—	183.3 ± 2.3	281.8 ± 39.5
PbHMPSP13	SPQ95810	320	100	PKc_like	Yes	No	76.9 ± 4.9	92.7 ± 10.0	117.6 ± 27.3	110.3 ± 10.0	201.7 ± 9.9	91.2 ± 8.4	73.5 ± 8.4
PbHMPSP14	SPQ96499	459	100	No	No	—	—	—	—	—	—	—	—
PbHMPSP15	SPQ96512	592	100	BTB/POZ	Yes	No	257.1 ± 12.7	—	—	—	—	130.9 ± 18.3	119.1 ± 4.5
PbHMPSP16	SPQ97596	374	100	Abhydro_lipase	Yes	Yes	37.9 ± 7.3	345.9 ± 67.2	256.3 ± 92.5	43301.0 ± 5807.5	10791.7 ± 3426.6	265.6 ± 75.1	1187.2 ± 207.5
PbHMPSP17	SPQ97819	490	100	Peptidase_S28	Yes	Yes	73.1 ± 17.1	119.7 ± 26.2	36.5 ± 6.4	93.0 ± 17.4	85.8 ± 15.0	63.7 ± 14.9	124.6 ± 29.9
PbHMPSP18	SPQ98758	478	100	PK_Tyr_Ser-Thr	No	—	—	—	—	—	—	—	—
PbHMPSP19	SPQ98922	319	100	GH16_fungal_Lam16A_glucanase	Yes	No	510.8 ± 65.4	—	—	—	—	308.1 ± 24.0	2365.8 ± 474.3
PbHMPSP20	SPQ99009	347	100	ChtBD1	Yes	Yes	160.0 ± 12.4	—	—	—	—	216.0 ± 7.2	306.7 ± 19.1
PbHMPSP21	SPQ99216	414	100	No	No	—	—	—	—	—	—	—	—
PbHMPSP22	SPQ99777	338	100	No	Yes	Yes	45.2 ± 1.9	78.0 ± 10.3	142.5 ± 51.2	293.5 ± 53.0	167.0 ± 36.3	477.2 ± 57.3	31.1 ± 2.2
PbHMPSP23	SPQ99827	895	100	CE4_SF	Yes	—	—	—	—	—	—	—	—
PbHMPSP24	SPQ99850	308	100	LRR_8	Yes	No	245.0 ± 20.4	—	—	—	—	306.2 ± 46.7	967.8 ± 143.3
PbHMPSP25	SPQ99917	550	100	Peptidase_C1	Yes	No	78.8 ± 3.4	40.1 ± 3.1	26.6 ± 2.5	12.1 ± 2.5	144.6 ± 16.2	41.5 ± 2.3	33.0 ± 6.6
PbHMPSP26	SPR00105	327	100	No	Yes	No	36.5 ± 2.3	42.9 ± 5.5	82.2 ± 14.7	242.2 ± 11.3	274.6 ± 11.9	196.8 ± 42.1	19.7 ± 1.1
PbHMPSP27	SPR00206	460	100	Ank_2	Yes	No	236.8 ± 20.2	—	—	—	—	52.9 ± 9.8	703.9 ± 42.2
PbHMPSP28	SPR00984	393	100	No	Yes	No	66.7 ± 3.6	36.5 ± 2.8	17.3 ± 3.4	122.5 ± 18.8	99.0 ± 9.3	20.6 ± 6.6	106.0 ± 7.4
PbHMPSP29	SPR01030	685	100	Tyrosinase	Yes	—	—	—	—	—	—	—	—
PbHMPSP30	SPR01068	428	95.873	PHA03326	Yes	—	—	—	—	—	—	—	—
PbHMPSP31	SPR01115	332	91.293	Protein kinase	Yes	No	178.3 ± 19.8	—	—	—	—	34.1 ± 3.2	743.8 ± 27.8
PbHMPSP32	SPR01575	841	100	NDK	Yes	—	—	—	—	—	—	—	—
PbHMPSP33	SPR01739	315	94.134	LRR_8	Yes	No	82.5 ± 8.8	69.9 ± 6.1	1700.2 ± 165.9	8441.3 ± 489.4	1477.6 ± 69.1	34.8 ± 10.2	145.0 ± 21.5
PbHMPSP34	SPR01965	355	89.296	LRR_8	Yes	No	37.9 ± 5.5	29.4 ± 10.1	0.51 ± 0.1	0.35 ± 0.03	1.1 ± 0.04	84.2 ± 7.0	29.7 ± 2.5
PbHMPSP35	SPR02066	366	100	No	Yes	Yes	38.8 ± 6.1	72.5 ± 10.8	238.0 ± 55.3	2729.3 ± 343.3	504.0 ± 96.9	88.9 ± 20.7	142.0 ± 18.2

Note: “—”indicates no date. NPR1 is the receptor of SA. After the depolymerization of NPR1 is induced by SA, *NPR1* is transferred into the nucleus and activates the expression of disease-related (*PR*) genes [49,50]. To further understand the regulation of PbHMWSPs in the SA pathway, the expression levels of *NPR1* were quantified in *N. benthamiana* leaves expressing PbHMWSPs. The results showed that except for PbHMWSP1 and PbHMWSP16, the PbHMWSPs could inhibit the expression of the *NPR1* gene (Figure 4B). Among these PbHMWSPs, PbHMWSP34 showed the most obvious inhibitory effect on the expression of *NPR1*. The expression level of *NPR1* in *N. benthamiana* leaves expressing PbHMWSP34 was 29.4% that of control. These results showed that different PbHMWSPs had different functions in the regulation of the SA pathway, with PBHMWSP34 being one of the most powerfully negative regulatory factors.

## Data Availability

Data are contained within the article and Appendix A.

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
