# Peer review of "Identification and Characterization of High-Molecular-Weight Proteins Secreted by Plasmodiophora brassicae That Suppress Plant Immunity"

_jof, 2024, doi:10.3390/jof10070462_

Round 1

Reviewer 1 Report

The research is interesting and provides new possible effectors involved in the establishment of disease caused by Plasmidiophora brassicae. The methods used and the results achieved allow to state that new effectors were identified. The manuscript requires a format review to clarify the focus and results of the research. The size of the proteins studied, for example, must be well established. I think the discussion could be improved by informing that these are new effectors that have not yet been described with this function. 

Please, follow the instructions and suggestions in the comments inserted in the attached PDF document.

Author Response

Dear Reviewer,

Thank you very much for your review and valuable comments on my manuscript. To provide you with more detailed information, I have organized the specific content into an attachment, which is included with this message. Please refer to the attachment for detailed information.

Thank you again for your time and assistance!

Reviewer 2 Report

This work is a useful contribution to the field of clubroot disease research. Until now many of the researchers testing for virulence factors from the secretome of P. brassicae have focussed on the smaller proteins <450 or <300 amino acids, as these are more similar to the effectors of other pathogens. As an intracellular pathogen the larger secreted proteins of P. brassicae are just as capable of impacting host cellular functions. The authors have identified secreted proteins with larger sizes, shown that they can be secreted in yeast and begun characterising some of them by testing their function in N. bethamiana leaves. They show several larger secreted putative effectors may suppress host responses to PAMPs and host hormone signalling pathways, thus presenting several interesting candidate effectors for future studies.

The results of host gene expression of heterologous expression of P. brassicae proteins in N. benthamiana leaves could be improved by making log (base 2 or base 10) transformation of the data. This would be particularly useful for showing the down-regulation of some responses in Figure 4B and may obviate the need for axis breaks. Furthermore, the statistical comparison (Student’s t-test) assumes a normal distribution of the data, I would be interested to know if the data was closer to meeting this assumption after log transformation.

The conclusion section provides some interesting context for the functional domains found in the candidate proteins. However, the writing style in this section might be improved, some of the sentences are very short, it has a fragmentary staccato style.

Minor points

16 “molecules secreted proteins” is not grammatical, perhaps “secreted proteins with low molecular weight”

16/17 “have been reported”

19 “proteins” “that are highly”

26 “a marker gene of the JA”

26 Is it informative to give the percentage here?  It is the proportion of those tested rather than those identified

28 perhaps “several SA” rather than “all SA” as not ALL possible SA marker genes have been examined

29 “is the first to report” rather than “firstly”

30 typo “mechanisms”

35 “employ”

40 “secrete” “cells”

41 “interfere with” “infect the host”

44 “oomycetes”

45”the apoplast”

45/46 “are able to enter into”

47 Effectors are not defined by a size limit and there are RxLR effectors >300 aa could phrase “no larger than” as “typically smaller than”

48 “except small” perhaps “apart from small” or “in addition to small”

49 “there have been few studies made on these”

51 Plasmodiophora (P). brassicae no need for the (P.) here

54 perhaps “occupies” or “inhabits” rather than “parasitizes in”

65 “pathotype” not plural

73 “P. brassicae” typo ital

75 P. brassicae” typo ital

75 “genes encoding HMWSPs”

82 “P. brassicae” typo ital

83 “P. brassicae” typo ital

84 “P. brassicae” typo ital

88 Brassica (B.) napus ital no need for (B.)

89 “clubroot gall”

90 B. napus ital

93 Were spores quantified, what amount of spores were used for inoculations?

93 B. napus ital

93 Nicotiana (N.) benthamiana no need for (N.) here

95 “that are specifically”

99 “P. brassicae” typo ital

100 “regions were”

106 the spore isolate sequence in the Schwelm paper is e3 not e310

111 “P. brassicae” typo ital

112 / 113 99 “P. brassicae” typo

113 Perhaps the definition of “highly expressed” could be clarified, it reads as though a gene with 0 reads in resting spores and >0 FPKM in planta could be classified as “highly expressed” what was the lowest in planta fpkm value for the candidates?

118 “both series”

120 “P. brassicae” typo

131 “peptides”

141 mg

145 “infection” “B. napus” ital

150 “Agrobacterium (A.) tumefaciens” no need for (A.)

152 “dilution to a OD600 of 1, the A. tumefaciens

153 “at the 5-6 leaf”

161 “inoculation” instead of “penetration”

163 “changing”

444 www.mdpi.com/xxx/s1 could not access supplemental tables

165 “mL”

167 “keeping … submerged”

168 “mL”

170 “were”

172 “was recorded over 40 minutes”

173 “Student’s”

175 “Quantifying the” “Quantification of the”

179 “PR1a” and throughout italics for gene names

182 “is shown”

184-186 “mL” “mM”

192 t-test

193 “Student’s”

200 “a signal peptide”

205 “a length greater than”

199-211 e3 not e310

210 “B. rapa” ital

211-212 “B. napus” ital

212 “B. oleracea” ital

Figure 1: the isolate name is e3 not e310 , all proteins are macromolecules

221 e3 italics for species names

227 this sentence is not grammatically correct

231 “abnormally”

245 “grew”

Figure 2 Gene names have been changed from PbHMWSP to PbMPSP

247 Italics for species names

251 “plants”

252 “mature proteins” repeated

257 “treatment”

261 “not”

279 Figure 3 was there any significant difference in the baseline RLU for transient expression of the HMWSPs before the flg22 was added?

289 “resistance”

292 “expression of”

293 -305 italics for PR1a and other gene names

305 PbMSP34 = PbHMWSP34?

Table1 Gene names have been changed from PbHMWSP to PbMPSP, macromolecular secreted proteins is not a useful definition as all proteins are macromolecules

What comparison does the Identity % refer to?

Could the Functional domains column be made wider to improve readability?

PR1 and other gene names should be in italics for table 1

Since PbBSMT is the published name of one of these proteins perhaps it should be included in the table 1

310 “after the depolymerisation od NPR1 is induced by SA,”?

313-316 NPR1 italics

Gene names not consistent PbMSP1, 16 34

319 PBMSP34 = PbHMWSP34

Figure 4 Log transformation of data may improve the visualisation of changes in expression, especially down-regulation of PbMPSP34 (PbHMWSP34?). Could the font size of the asterisks indicating significance be increased

331 “research has”

336-346 PDF1.2 LOX italics

340,342 PbMSP25

347 PbMSP6

351 “gene”

360 “P. brassicae” typo ital

363-373 ERF italics

368 PbMSP26

371-372 PbMSP19

396 “negatively”

397 “pathways”

416 “There are some lipid droplets in P. brassicae [63]” – expand

430 “Consequently” non-sequitur

432 kinases have many functions outside of immune signalling as well

Author Response

(The authors gave the same response as above.)

Reviewer 3 Report

The authors in their manuscript entitled “Identification and characterization of high molecular weight proteins secreted by Plasmodiophora brassicae that suppress plant immunity” present the effects of overexpression of selected HMW proteins in the expression of various marker genes related to hormone-mediated plant responses to pathogens.

The methodology and assays applied follow a standard approach are the implementation is correct. My main question and comment are regarding the significance of the conclusions stated in relation to certain experimental results.  Other comments regarding technical issues are also presented.

In specific:

1.      The authors in their title state that there is suppression of plant immunity. This conclusion is overestimated in my opinion since the deduced results refer only to alterations in expression of certain marker genes. This does not necessarily prove suppression of plant immunity. There are no relevant assays that measure the plant responses to infection of Plasmodiophora brassicae and specifically in roots which is the actual site of infection, however, nor to other pathogens that infect other plant organs (e.g. leaves). To conclude that there is suppression of plant immunity, studies with the relevant gene mutants would be the best approach.

2.      In relation to the above, the authors choose to study the roles of the secreted HMW proteins practically by applying ectopic overexpression of these in leaves of another plant (N. benthamiana). This approach, though technically correct, cannot necessarily assign immunity suppressive roles to the proteins. I will try to be more precise.

a.      The selected cDNAs are cloned in an expression vector under a 35S promoter which by default over-expresses the proteins. The relative protein amounts produced are in excess when compared to the natural infection status in roots. Any effect of the proteins can be assigned only to their over-expression, and not necessarily to their native roles. I agree that there is an effect on the expression of certain marker genes, however, this can be assigned to the over-expression of the proteins.

b.     The effect is studied in leaves and actually in a heterologous system. So, any conclusions should be drawn from this point of view. The expression of the hormone-related marker genes is studied only in tobacco plants. A parallel study of the corresponding gene in roots (primarily) and leaves (also) in Brassica plants under normal infection conditions could give an indication that could be related-discussed with the results from tobacco plants.

3.      L.102; please correct the phrase in relation to the “signal peptide” and “transmembrane regions”.

4.      L. 214; The authors refer to “tissues”. Does this refer to “samples”? If so please correct, if not the please provide additional information regarding the type of tissue the genes were studied in.

5.      L. 231; The authors state that there is “unnormal” growth of certain strains, however, the corresponding images in figure 2 so just a slower growth rather than unnormal. Please revise and comment on this in the manuscript accordingly.

6.      L. 252; Please provide the sequence entries for the cloned genes deposited in the relevant database. There is no reference to this in any part of the manuscript. Please, make sure that the relevant IDs for the cloned genes appear in the manuscript.

7.      In figure legends of figures 4-6 please provide details for the abbreviated names used.

8.      I would suggest that the results and discussion parts and the relative conclusions are further elaborated, regarding the significance of the results. I agree that there is a connection between the over-expression of the proteins and hormone-related marker genes, and this should be promoted. However, the system used by default shows these effects ectopically (leaves), in a heterologous system (tobacco) and under “forced” conditions (overexpression). The latter (overexpression), which is the correct description of the state, is actually reported only once in the manuscript (I spotted only line 280). The presentation of the results under this point of view I feel that would attribute the correct significance of the results and would enlighten the research work.

Author Response

(The authors gave the same response as above.)

Reviewer 4 Report

The work by Feng and colleagues titled "Identification and characterization of high molecular weight proteins secreted by Plasmodiophora brassicae that suppress plant immunity" is a very comprehensive investigation with interesting results. The study identifies 35 proteins through transcriptomic techniques and in silico predictions in P. brassicae with a potential function or role in modulating plant immunity, akin to effector proteins. Other cloning and expression experiments successfully confirmed the potential function of some of these proteins, such as PbHMWSP34 27, which regulates the JA (LOX), ET (ERF), and SA (PR1a, PR1b, PR2b, PR5, NPR1) signaling pathway marker genes.

However, it appears that the manuscript contains various English errors that should be corrected by a native editor or some professional service, as the Introduction also requires fluency. The reading experience is rather abrupt.

I only have a few minor comments:

Line 30: correct "mechinisms" to "mechanisms." Change any keywords that are not in the title of the work. For example, replace "high molecular weight secreted protein; Plasmodiophora brassicae; plant immunity." Please use commas in lines 35-37, first paragraph of the Introduction. Mention a reference that discusses N. benthamiana and its interaction with... Line 49, please add a reference: "…while there is still few study on these HMWSPs." Check English in Line 58: The subject "genome" is singular, so "was" should be used instead of "were." Line 73, italicize "P. brassicae" and correct it. Also, in the same line, "the" should be used before "P. brassicae genome" to indicate specificity. Line 75, italicize "P. brassicae." Lines 82, 83, and 84, italicize "P. brassicae." Line 88, italicize "Brassica (B.) napus." Correct Figure 1 legends. Italicize scientific names. Optional: Adjust the Figures of relative expression with the same colors, just for aesthetic reasons. Because each figure is in a different color. For example, all in grayscale and some with shades of gray or another color preferred by the authors.

Author Response

(The authors gave the same response as above.)

Round 2

Reviewer 3 Report

I thank the authors for their replies. I understand their rationale regarding the overexpressing system used and the amount of work needed in a relevant experiment using A. thaliana. However, this would not change much since this is also a heterologous system. As the authors state in their replies their work is a preliminary part of their concept work. Though their replies explain the logic behind the experiments (which I follow), my impression is that the conclusions in the form they are presented in the manuscript overestimate the essence of the results. As it was stated in the first review run there is a clear connection regarding alterations in expression of certain marker genes, but this does not necessarily prove suppression of plant immunity.

I would suggest that the Discussion part is elaborated more (this was an initial comment as well) and in a way that would highlight the actual significance of the results. The authors state in the discussion that “The majority of the P. brassicae PbHWMSPs reported in this paper could suppress plant immunity”; this means that there is an indication, not a proof that there is suppression. There are alterations in gene expression, however the two concepts are different. I feel that this should be reflected in the manuscript title as well where the word suppression guides the reader to a certain concept. Please highlight the relevant changes in the 3rd version of the manuscript to be able to follow the changes.

Thank you.

Detailed comments are included in the same paragraph as the major ones. At this stage I feel that a structural reform of the discussion part and certain changes regarding concepts could improove the manuscript and promote its essence.

Round 3

Reviewer 3 Report

No further comments

No further comments